# Antioxidant Systematic Alteration Was Responsible for Injuries Inflicted on the Marine Blue Mussel *Mytilus edulis* Following Strontium Exposure

**DOI:** 10.3390/antiox13040464

**Published:** 2024-04-14

**Authors:** Zihua Cheng, Mengxue Xu, Qiyue Cao, Wendan Chi, Sai Cao, Zhongyuan Zhou, You Wang

**Affiliations:** 1College of Marien Life Sciences, Ocean University of China, Qingdao 266000, China; chengzihua@stu.ouc.edu.cn (Z.C.); caoqiyue@stu.ouc.edu.cn (Q.C.); caosai@ouc.edu.cn (S.C.); zhouzhongyuan@ouc.edu.cn (Z.Z.); 2Laboratory for Marine Ecology and Environmental Science, Qingdao Marine Science and Technology Center, Laoshan Laboratory, Qingdao 266200, China; 3Marine Science Research Institute of Shandong Province, Qingdao 266100, China; wyf9768@stu.ouc.edu.cn (M.X.); chiwendan01@shandong.cn (W.C.); 4Qingdao Key Laboratory of Coastal Ecological Restoration and Security, Qingdao 266100, China

**Keywords:** nuclear-contaminated water, stable strontium, *Mytilus edulis*, toxic effect, risk assessment

## Abstract

The ionic properties of strontium (Sr), a significant artificial radionuclide in the marine environment, were estimated using a stable nuclide-substituting experimental system under controlled laboratory conditions. The bio-accumulation of Sr and its impacts, as well as any possible hidden mechanisms, were evaluated based on the physiological alterations of the sentinel blue mussel *Mytilus edulis*. The mussels were exposed to a series of stress-inducing concentrations, with the highest solubility being 0.2 g/L. No acute lethality was observed during the experiment, but sublethal damage was evident. Sr accumulated in a tissue-specific way, and hemolymph was the target, with the highest accumulating concentration being 64.46 µg/g wet weight (ww). At the molecular level, increases in the levels of reactive oxygen species (ROS) and malondialdehyde (MDA) and changes in ROS components (H_2_O_2_, O^2−^, and -OH) and antioxidant system activity indicated that the redox equilibrium state in hemocytes was disturbed. Furthermore, the rise in the hemocyte micronucleus (MN) rate (4‰ in the high-concentration group) implied DNA damage. At the cellular level, the structures of hemocytes were damaged, especially with respect to lysosomes, which play a crucial role in phagocytosis. Lysosomal membrane stability (LMS) was also affected, and both acid phosphatase (ACP) and alkaline phosphatase (AKP) activities were reduced, resulting in a significant decline in phagocytosis. The hemolymph population structure at the organ level was disturbed, with large changes in hemocyte number and mortality rate, along with changes in component ratios. These toxic effects were evaluated by employing the adverse outcome pathway (AOP) framework. The results suggested that the disruption of intracellular redox homeostasis is a possible explanation for Sr-induced toxicity in *M. edulis*.

## 1. Introduction

On 24 August 2023, Japan initiated a plan to discharge nuclear-contaminated water from the Fukushima Disaster (2011) into the Pacific Ocean, which attracted worldwide attention with regard to potential impacts on marine ecosystems. As early as 2012, German scholars demonstrated that nuclear-contaminated water was spreading through ocean currents and would affect most of the Pacific Ocean within 59 days and the whole ocean within 10 years [1]. For instance, a survey conducted in 2012 discovered elevated levels of ^90^Sr in the Pacific Northwest region [2].

Nuclear-contaminated water contains over 60 types of radionuclides, 8 of which are assumed to pose extreme threats to the marine environment: ^3^H, ^134^Cs, ^137^Cs, ^90^Sr, ^89^Sr, ^60^Co, ^129^I, ^125^Sb, and ^106^Ru. These radionuclides are the characteristic nuclides of nuclear accidents. Marine organisms act as vectors for radionuclides in seawater. The radionuclides attach to the surfaces of these organisms’ bodies or are ingested and subsequently undergo biotransmission and bioaccumulation along the food chain. As humans are positioned at the apex of the food chain and rely on a multitude of ocean resources, the release of nuclear-contaminated water poses an inevitable threat to human health and safety due to the transportation of radionuclides.

The hazards of radionuclides with respect to living organisms can be classified as radiotoxicity and ionic toxicity. Radiation attacks the hematopoietic system, nervous system, and immune system of organisms and causes acute damage [3,4]. Certain radioactive substances with long half-lives (such as ^137^Cs and ^90^Sr) can accumulate in living organisms and cause chronic damage by obstructing the synthesis of proteins and DNA [5]. The ionic toxicity of certain radionuclides is more harmful to the environment than their radioactivity. The accumulation of heavy metals in organisms can lead to abnormal growth, shortened lifespans, and increased mortality [6]. Humans occupy the highest position in the food chain and are susceptible to more serious health hazards from consuming large quantities of fish and shellfish. Damage to human DNA caused by radionuclides can lead to premature fatalities, deformities, disabilities, and cancers. Moreover, such damage is transgenerational, and genetic harm can persist for thousands of years. A search of the literature using the keyword “marine organisms” uncovered few studies on the marine ecological effects of characteristic nuclides. Research on nuclear-contaminated water has mainly focused on assessing nuclide transport and diffusion, radionuclide monitoring, and radioactive source analysis. Therefore, it is essential to investigate the marine biological effects of radionuclides to comprehend the present state of nuclear pollution in marine environments and assess the influence of nuclear facilities and the aftermath of nuclear accidents.

The benthic environment is the primary location for nuclide accumulation. Radionuclides can be deposited on the seafloor in biological carcasses or excreta, and the release of the resulting sediments can cause secondary pollution in the marine environment and organisms. Mussels are benthic organisms, and the relationship between mussels and sediments is complex and interdependent. Mussels have been widely used as pollution-monitoring organisms and as important experimental models in ecological studies [7,8]. Therefore, we used the sentinel blue mussel *Mytilus edulis* as a study object and centered our research on strontium (Sr) in nuclear-contaminated water. Immunity is an important response pathway with respect to environmental stresses, and hemocytes play a critical role in the bivalve immune system [9]. Hemocytes are also responsible for essential physiological functions, including nutrient transport, digestion, and shell and tissue formation, as well as translocation, excretion, self-repair, and maintenance of homeostasis [10]. Therefore, in our study, we prioritized examining the immune functions of hemocytes in mussels to elucidate the impact of Sr on mussels.

The adverse outcome pathway (AOP), a conceptual framework introduced in 2010, permits the assessment of toxic effects at different organizational levels by integrating multiple forms of toxicological information [11]. In 2012, the Organization for Economic Cooperation and Development (OECD) launched a program for developing AOPs in the form of analytical constructs that describe a series of linked events that are causally related and lead to an adverse health/environmental effect [12,13]. The AOP concept is currently used in the regulatory assessment of chemical risks, the safety assessment of industrial chemicals, the discovery and development of new products, and the testing of pharmaceutical products’ health and environmental quality. We have comprehensively described an AOP of 2,2′,4,4′-tetrabromodiphenyl ether (BDE-47) with respect to *Brachionus plicatilis*. Therefore, in the present study, we assessed the acute and subacute toxic effects of Sr in mussels with the support of the AOP framework by examining the bioconcentration of exposure in different tissues and investigating related impairments in immune function within mussel hemocytes. This study aids our understanding of the detrimental impacts of Sr on marine ecosystems and provides a significant theoretical basis for ecological risk assessment using mussels. Additionally, this study offers new perspectives on the ecological consequences of nuclear-contaminated water.

## 2. Materials and Methods

### 2.1. Cultivation of Mussels

Mature *M. edulis* individuals (over 1 year old; shell length: 5.4 ± 0.5 cm; wet weight: 7.03 ± 0.78 g) were collected from Laoshan Bay, Shandong Province, China (36°15′ N, 120°40′ E), and transferred to our laboratory for maintenance. The mussels were acclimated in tanks containing 75 L of natural seawater. The mussels were fed the microalgae *Platymonas helgolandica* (1.5 × 10^5^ cells/mL) once daily and allowed to acclimate for 7 days under the following conditions: storage in seawater (renewed daily) with constant aeration, salinity of 31 ± 1.0, pH 8.1 ± 0.1, temperature of 18 ± 2 °C, and a 12/12 h light/dark cycle.

### 2.2. Experimental Setup

#### 2.2.1. Simulation of Sr Exposure

Stable nuclides and radioactive nuclides possess identical chemical and biological properties. Therefore, stable nuclides are frequently used as substitutes for radioactive compounds in studies of ionic toxicity [14,15]. The mussels in this study have a relatively short life cycle, and the decay effect of ^90^Sr throughout their life cycle is insignificant. Therefore, there was no need to adjust the experimental data using decay correction. For this study, SrCl_2_ (AR, purity > 99.5% S817918, MACKLIN, Shanghai, China) was selected as the source of stable Sr, and the maximum solubility of SrCl_2_ in seawater was determined to be 0.2 g/L. Preliminary acute toxicity experiments indicated that the highest solubility level of strontium in seawater was not acutely lethal for mussels. However, it did hinder the normal physiological responses of the mussels. The criterion for individual mortality was the inability to respond when the mantle of the mussel was touched [16]. The reference Sr content in some sea areas is 0.002 g/L [17], and the concentrations of SrCl_2_ in the subsequent experiments on sublethal toxicity were 0.2 g/L (high-concentration group), 0.02 g/L (medium-concentration group), and 0.002 g/L (low-concentration group). The water volume utilized in this experiment was 10 L. The Sr concentration of the water was adjusted to meet the experimental requirements by adding 2, 0.2, and 0.02 g of SrCl_2_ to three different experimental groups. Natural seawater was used in the control group. Thirty healthy adult mussels were used in each group, with an approximate density of 330 mL/mussel. The experimental period was 21 d, and it was preceded by 7 d of fasting. The experimental mussels were fed microalgae daily. Additionally, the seawater containing the indicated concentration of Sr was replaced after 2 h of feeding.

#### 2.2.2. Bioaccumulation of Sr in *M. edulis*

Samples of hemocytes, digestive glands, gonads, and gills were collected on the 21st day of the experiment. The collected samples were snap-frozen in liquid nitrogen for 10 min and then stored at −80 °C until analysis. All tissue samples were weighed (with an approximate 0.5 g wet weight (ww) as a minimum sample mass) and separately transferred into digestion flasks. Then, 10 mL of HNO_3_ (MOS) was added to the samples, which were digested in an automatic digestion apparatus (ST-60, POLYTECH, Beijing, China). The digestive procedure consisted of 120 °C for 1 h, 140 °C for 1 h, 160 °C for 1 h, and 180 °C for 45 min. After digestion, acid was eliminated from the samples, and the samples were diluted to 10 mL with high-purity water. Sr concentrations were measured using inductively coupled plasma–atomic emission spectrometry (ICP–AES, SPECTRO ARCOS EOP, SPECTRO Analytical Instruments GmbH, Kleve, Germany) and expressed as μg/g ww.

#### 2.2.3. Effects of Sr on Hemolymph Composition in *M. edulis*

The total number of circulating hemocytes in each milliliter of hemolymph was determined by measuring the total hemocyte count (THC). Twenty microliters of pretreated hemolymph was fixed with Bayer’s fixative (containing 2% NaCl and 4% paraformaldehyde) and diluted in 1:3 ratio, and the hemocytes were counted using a cell counter under a 40× light microscope.

Flow cytometry (FCM) analysis was used to determine changes in hemocyte composition and phagocytosis (the testing standards are shown in the Appendix A). According to cell size and subcellular complexity, hemocytes can be classified as hyaline cells or granular cells [18]. The proportion changes of hemocytes were verified using FCM analysis according to a method derived in our laboratory [19]. The mortality of hemocytes was detected with FCM analysis according to the method reported by Hégaret et al. [20]. Phagocytosis is the primary immune defense mechanism of hemocytes against xenobiotics and pathogens. The changes in phagocytosis were determined using FCM analysis by testing the number of ingested fluorescent microspheres [19,21].

#### 2.2.4. Effects of Sr on Hemocyte Structure and Function in *M. edulis*

The collected hemocytes without anticoagulant were spiked with glutaraldehyde until reaching a final concentration of 5% and incubated overnight at 4 °C until the cells coagulated into clusters. After cryo-agar embedding, staining, dehydration treatment, and sectioning, the hemocytes were observed using a transmission electron microscope.

Lysosomes are important providers of immune function for granulocytes, and changes in their function are reflected via LMS and the activities of related enzymes. LMS analysis was performed by determining the neutral red retention time (NRRT) according to the method devised by Regoli et al. [22]. ACP and AKP are important lysosomal hydrolases. Their phosphatase activities were determined using the disodium phenyl phosphate colorimetric determination method devised by Hervio et al. [23].

Five mussels were randomly selected from each group on the 21st day of the experiment. Fifty microliters of hemolymph without anticoagulant was taken and spread evenly on slides precoated with 10% polylysine (Sigma, St. Louis, MO, USA). The slides were air-dried for 30 min and then fixed in absolute methanol (BDH, Poole, UK) for 15 min. The slides were stained with 5% Giemsa staining solution (BDH, Poole, UK) for 20 min, air-dried, and fixed with coverslips with the aid of DPX. MN was evaluated using a microscope. At least 1000 cells were scored from each slide (three slides per group of mussels) [24]. The lymphocyte MN rate (‰) was calculated as the number of micronucleate cells present per 1000 cells.

#### 2.2.5. Effects of Sr on Oxidation–Reduction Homeostasis and Antioxidant System Activity in *M. edulis*

The intracellular ROS content of hemocytes was determined using the DCTH-DA assay [25]. ROS content was expressed as the fluorescence value of 2 × 10^6^ cells. MDA is a potent bioindicator of oxidative stress and can reflect the degree of lipid peroxidation in an organism. MDA content was determined using the TBA (thiobarbituric acid) method, with a maximum absorption peak at 532 nm. Changes in ROS components and the activity of the antioxidant system in hemocytes were evaluated using Nanjing Jiancheng kits. The ROS components included H_2_O_2_, O^2−^, and -OH. Indicators related to the antioxidant system included catalase (CAT), superoxide dismutase (SOD), glutathione peroxidase (GPX), glutathione reductase (GR), and reduced glutathione/oxidized glutathione (GSH/GSSG).

### 2.3. Statistical Analysis

To calculate the mean and standard deviation, five different sample replicates (*n* = 5) were used. Graphs were obtained using SigmaPlot 12.5 analysis. For the experimental and control groups, one-way ANOVA was carried out using SPSS 22.0 software with a significance threshold of *p* < 0.05; *p* < 0.01 was considered highly significant. The impact of concentration variables on each parameter was investigated via univariate ANOVA using a generalized linear model (GLM). FlowJo v10.8.1was utilized to analyze the results of FCM analysis. Nonparametric tests (Kruskal–Wallis test) were employed to assess the impacts of differences in exposure time and concentration. Data statistics were acquired using Excel 2019, and data were analyzed using Origin 8.5.

## 3. Results

### 3.1. Bioaccumulation of Sr in M. edulis

The levels of Sr bioaccumulation in different tissues of *M. edulis* after the 21-day Sr exposure period are shown in Figure 1. The levels of Sr bioaccumulation in hemocytes, digestive glands, gonads, and gills were higher in the medium-concentration and high-concentration groups than in the control group and depended on the concentration of Sr in the seawater (Figure 1A). In the high-concentration group (0.2 g/L), the levels of Sr bioaccumulation in hemocytes, digestive glands, gonads, and gills were 64.46 µg/g ww, 48.69 µg/g ww, 48.78 µg/g ww, and 58.23 µg/g ww, respectively. The bioaccumulation of Sr in mussels was tissue-specific and typically highest in hemocytes (Figure 1B).

### 3.2. Effects of Sr on Hemolymph Composition in M. edulis

As shown in Figure 2A, at the beginning of Sr exposure, the hemocyte counts were significantly elevated in the low-concentration group and obviously decreased in the medium-concentration group and high-concentration group compared with the control. The hemocyte number continued to increase with prolonged exposure and higher Sr concentrations. Hemocyte mortality increased and then decreased upon increasing exposure concentration (Figure 2B). Sr exposure had no significant effect on the percentage of granulocytes in the total hemocytes, although the numbers of granulocytes decreased slightly in the high-concentration group (Figure 2C). The above results suggested that Sr altered hemolymph composition.

### 3.3. Effects of Sr on Hemocyte Structure and Function in M. edulis

The granular hemocytes were observed using a transmission electron microscope. After 21 days of Sr exposure in the low-concentration group, the granules in mussel granulocytes increased in size and decreased in number, and lysosomes, rough endoplasmic reticula, and mitochondria became more numerous (Figure 3B). There were even fewer granules in the granulocytes in the medium-concentration group (Figure 3C). In the high-concentration group, most of the granules disappeared; there were gaps between the granules and the outer membrane, with obvious vacuolization, and the nuclear membrane became thinner and even degraded and disappeared (Figure 3D). The above results indicated that Sr destroyed the cellular structures of mussel hemocytes and damaged their immune function.

As shown in Figure 3E, the LMS of mussel hemocytes changed significantly following Sr exposure. LMS was significantly higher in all the experimental groups than in the control group (*p* < 0.01) on day 10 but not day 21. ACP activity decreased throughout Sr exposure (Figure 3F). Consistent with the changes in LMS, AKP activity increased at the beginning of exposure and then decreased until the end of exposure (Figure 3G). Phagocytosis in mussel granulocytes was evaluated on day 21. The phagocytosis of hemocytes was significantly lower in the medium-concentration group and high-concentration group than in the control group (*p* < 0.05) (Figure 3H). These results suggested that Sr caused lysosomal perturbation in mussel hemocytes and impaired their phagocytosis.

The MN contains one or more nucleosomes and is independent of the main nucleus in the cell. The volume of the MN is approximately one-third that of the normal nucleus. As depicted in Figure 3I, the morphology of mussel hemocytes was normal in the control group, low-concentration group, and medium-concentration group, and no MNs were found. The high-concentration group had an MN frequency of approximately 4‰, with the occurrence of binucleated (Figure 3J) and multinucleated (Figure 3K) cells. These findings suggest that exposure to high concentrations of Sr increased the incidence of micronuclei compared to the control group (*p* < 0.05) and disrupted hemocyte nucleus formation.

### 3.4. Effects of Sr on Oxidation–Reduction Homeostasis and Antioxidant System Activity in M. edulis

As shown in Figure 4A, on day 10 of exposure, the ROS levels in the mussel hemocytes increased significantly when increasing the Sr concentration. On day 21, ROS levels were significantly lower in all the experimental groups compared to the control group. On day 21, changes in three ROS components (H_2_O_2_, O^2−^, and -OH) were examined in mussel hemocytes. Overall, H_2_O_2_ content increased significantly with increasing exposure concentration (Figure 4B). O^2−^ content was significantly higher in the low-concentration group than in the other experimental groups (Figure 4C). Compared with the control group, -OH content decreased significantly in the medium-concentration group and high-concentration group and was lowest in the high-concentration group (Figure 4D). The changes in MDA content were consistent with the changes in ROS; MDA content increased and then decreased with an increasing exposure time (Figure 4E). GSH/GSSG was higher in the high-concentration group (Figure 4F). The above results indicated that Sr exposure disrupted the redox homeostasis of mussels. On day 21 of Sr exposure, changes in the antioxidant system of mussel hemocytes were examined. Compared with the control, CAT activity was not significantly different in the low-concentration group and medium-concentration group but increased significantly in the high-concentration group (Figure 4G; one-way ANOVA, *p* < 0.01). SOD activity did not change significantly but decreased and then increased with an increasing exposure concentration (Figure 4H). GPX activity changed slightly in the low-concentration group and medium-concentration group but decreased significantly in the high-concentration group (Figure 4I; one-way ANOVA, *p* < 0.05). GR activity decreased significantly in all treatment groups (Figure 4J). The above results indicate that Sr exposure affected the activity of the antioxidant system in the mussels.

## 4. Discussion

Mussels are typical marine *r*-strategy taxa with a distinct sensitivity to environmental pollution. Because the half-life of ^90^Sr is approximately 28 years, the impacts of its β rays could be ignored during the 21-day experimental period in the present study. Instead, this study focused on the chemical properties of the heavy metal Sr itself. Taken together with our previous research on the stable nuclide ^133^Cs [26], our results suggest that substituting stable nuclides is a generalizable approach. The stable nuclide substitution method provides a solution to the limitations and high risks posed by radionuclide studies. Furthermore, it simplifies the exploration of marine ecological effects and the risk assessment of nuclides commonly associated with nuclear accidents.

The bioaccumulation of metals in aquatic organisms, specifically bivalves, can signal the potential for metal contamination in aquatic environments [27,28]. In this study, the bioaccumulation of Sr in the hemocytes, digestive glands, gonads, and gills of mussels increased significantly with increasing concentrations of Sr exposure in the medium-concentration group and high-concentration group. These results are consistent with the bioaccumulation of other heavy metals in freshwater mussels reported previously [29]. Shellfish and other aquatic animals can ingest metals through two pathways: absorbing dissolved metal ions through their gills or acquiring particulate heavy metals in food through their digestive glands [30,31]. The digestive glands act as temporary storage organs for metals, and bioconcentration in these glands may only indicate short-term environmental contamination [32]. The results showed that bioaccumulation was consistent across all the concentration/treatment groups. The most pronounced bioaccumulation of Sr was observed in hemocytes, followed by gills, gonads, and digestive glands. These findings suggest that hemocytes may be the primary target tissue for Sr bioaccumulation and that hemolymph may play a role in redistributing or excreting Sr from organisms into the surrounding water [33]. Furthermore, this study was carried out during the nonreproductive phase of mussels, resulting in low gonad bioaccumulation. It is possible that the tissue bioaccumulation profile may change during the reproductive phase, with the gonads potentially becoming a primary, impacting reproductive behavior. We plan to conduct further investigations to examine the toxicological mechanisms of pollution stress during the reproductive phase of mussels.

Hemocyte morphology, total hemocyte counts, and differential hemocyte counts are crucial parameters for evaluating the immune competency of bivalves [34,35,36]. We investigated the hemolymph compositions and observed a marked rise in hemocyte counts during the initial stage of exposure to low concentrations of Sr, suggesting that mussels exhibit a stress response when subjected to external stress. As the exposure concentration and time increased, the hemocyte count increased, and the cell death rate first increased and then decreased. This may indicate compensation for impaired immune function in the organism. Stress also affected the proportion of granulocytes in the hemolymph. The findings indicate that exposure to Sr modifies the hemocyte composition present in the hemolymph, primarily influencing granulocytes and ultimately impeding the hemolymph immune response.

The maintenance of normal function requires cellular structure integrity. A clearer view of hemocyte morphology and immune status assessment in bivalves can be achieved through transmission electron microscopy [37,38]. The present study revealed that the cell membranes and nuclear membranes of hemocytes were smooth and intact under normal growth conditions. Exposure to high Sr concentrations induced considerable granulocyte damage, including significant vacuolization, swelling, thinning of the cell and nuclear membranes, lysosomal lysis, and chromatin condensation. This indicates that exposure to Sr disturbs hemocyte composition and harms multiple organelles. Lysosomes are crucial organelles that are involved in hemocyte phagocytosis, and we found that LMS increased upon initial exposure to Sr and then decreased. Organismal stress initiates lysosomal damage and ultimately leads to a decrease in LMS. To cope with environmental stress, lysosomes utilize important nonspecific hydrolases, such as ACP and AKP, to break down foreign substances. Our study demonstrated that exposure to Sr significantly reduced the activities of ACP and AKP in mussels, indicating lysosomal disorders. These changes impeded phagocytosis of hemocytes and suppressed immune function in mussels. Hemocyte micronucleus assays have been widely applied in studies of the genetic toxicology of freshwater and marine mussels [39,40,41]. Hemocyte MN frequency was elevated at the high concentration of Sr exposure, thus confirming prior findings that exposure to genetic toxicants increases the incidence of micronucleate hemocytes in mussels. No mussel loss or mortality occurred following 21 days of continuous exposure to Sr. This implies that the genotoxic effects are measurable through MN analysis and that MN frequency may indicate damage to DNA. However, it is important to note that the MN frequency did not differ significantly between the low- and medium-concentration groups. This suggests that the assay may not be appropriate for biomonitoring studies because it may incorrectly indicate the absence of genotoxicity at low exposure concentrations [24]. Furthermore, the frequencies of MNs in the experimental groups were no more than three times higher than those in the control group, indicating a possible limited ability of bivalve hemocytes to induce MN formation and the presence of a plateau or saturation effect [41]. In conclusion, exposure to Sr has considerable toxic effects on mussel hemocytes. These effects include elevated MN frequency and disruption of hemocyte nucleation, without taking into account β-rays. This also implies that exposure to Sr might impact transcription in hemocytes and thus affect gene function and immune function, among other effects.

External environmental stress can disturb intracellular ROS metabolism, resulting in ROS overproduction. The three primary ROS components produced by organisms are H_2_O_2_, O^2−^, and -OH. MDA, a typical byproduct of membrane lipid peroxidation, indicates the degree of organismal lipid peroxidation and the extent of cellular damage [42]. To counter the overproduction of ROS, organisms utilize two defense mechanisms, namely, enzymatic and nonenzymatic systems. The enzymatic defense system employs SOD, CAT, GPX, and GR as essential protective enzymes, and the nonenzymatic defense system relies on GSH as an important antioxidant [43]. The GSH/GSSG ratio can indicate changes in cellular redox homeostasis. Abnormal ROS metabolism and impaired antioxidant system function can cause damage to the cell membrane system, impacting normal cellular function and even leading to cell death [44]. We found that ROS levels and antioxidant enzyme activity were increased in mussel hemocytes during the early stages of exposure to Sr, resulting in membrane lipid peroxidation. The decline in ROS levels in the later stages of exposure may be attributed to the suppression of the overall metabolic activity of the mussels. Simultaneously, exposure to Sr severely impaired mussel immune function. The increase and subsequent decrease in membrane lipid peroxidation also confirmed the production of an immune stress response by the organisms in the early stage of stress. Although the changes in pivotal antioxidant enzymes varied, they were strongly correlated with the changes in ROS components. In the high-concentration group, CAT activity increased significantly and was positively correlated with H_2_O_2_ levels. This implied that exposure to Sr promoted the generation of H_2_O_2_ and, in turn, CAT activity. SOD may respond indirectly to an organism’s ability to eliminate O^2−^. O^2−^ levels increased in hemocytes, indicating an impact of Sr exposure on intracellular redox homeostasis. In the low-concentration group, GPX may have eliminated -OH, resulting in a minimal increase in -OH levels, whereas H_2_O_2_ levels remained relatively high. However, in the high-concentration group, despite the suppression of GR activity along with a decrease in GPX activity, the GSH/GSSG ratio remained high. In conclusion, exposure to Sr affected the production of ROS components. It also affected the activity of the antioxidant system in mussels and disrupted redox homeostasis in hemocytes.

We used the AOP framework to integrate these results and understand the toxic impact of Sr exposure on mussels at the molecular, cellular/tissue, organ, and organism levels (Figure 5). We concluded that Sr exposure disrupts redox homeostasis by inducing ROS production and causing hemocyte damage in blue mussels. This exposure suppresses mussel activity by disrupting hemocyte structure and immune function and altering hemolymph composition. Exposure to the highest solubility level of Sr (0.2 g/L) did not cause mussel death but significantly suppressed their activity, as evidenced by slow opening and closure and delayed retraction or closure of the mouth when the mantle was lightly stimulated. Taken together, our findings indicate that the release of nuclides in the discharge of nuclear-contaminated water may cause severe damage to marine organisms and their ecosystems. Furthermore, Sr and Ca have similar chemical properties [45], and nuclides can substitute for essential elements and harm organisms; thus, the toxic effects of Sr may be closely linked to various physiological processes that involve Ca ions. For instance, invertebrates rely on hemolymph coagulation as a critical humoral immunity mechanism, and Ca^2+^ is an indispensable coagulation factor. Additionally, Ca^2+^ participates in biological signaling, regulates numerous enzymes, and modulates immune system activity. This study prompted a follow-up investigation into the biological effects of Sr stress.

## 5. Conclusions

The present study evaluated the toxic effects of Sr by using the AOP framework. Sr tends to accumulate in the hemocytes, gills, digestive glands, and gonads of mussels, with higher concentrations in hemocytes. Exposure to Sr disrupts hemocyte redox homeostasis, induces DNA damage, and adversely affects cellular structure, resulting in disruption of the immune function of hemocytes. Exposure also affects the composition of hemolymph, a vital immune organ, ultimately impairing mussel health. This study not only demonstrates that Sr harms marine mussels and endangers ecosystems but also offers a dependable research model for assessing ecological impacts.

## Figures and Tables

**Figure 1 antioxidants-13-00464-f001:**
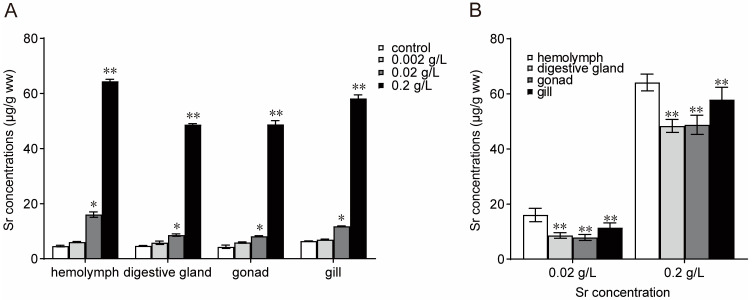
Sr concentrations in the main tissues of *M. edulis* over the Sr exposure period. (**A**): Differences between different concentrations of Sr exposure; (**B**): Differences between different tissues. * Significant differences at *p* < 0.05 level; ** Significant differences at *p* < 0.01 level.

**Figure 2 antioxidants-13-00464-f002:**
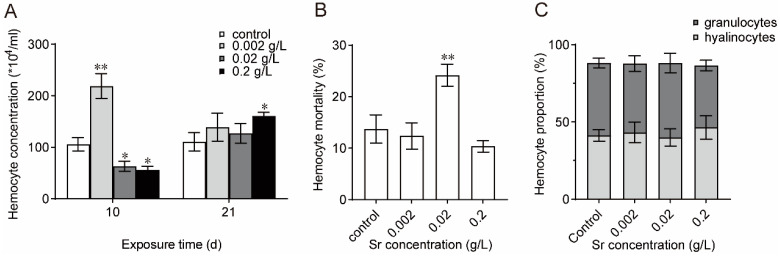
Changes in hemolymph community induced by Sr (*n* = 5). (**A**): THC; (**B**): Mortality; (**C**): Hemocyte proportion. * Significant differences at *p* < 0.05 level; ** Significant differences at *p* < 0.01 level.

**Figure 3 antioxidants-13-00464-f003:**
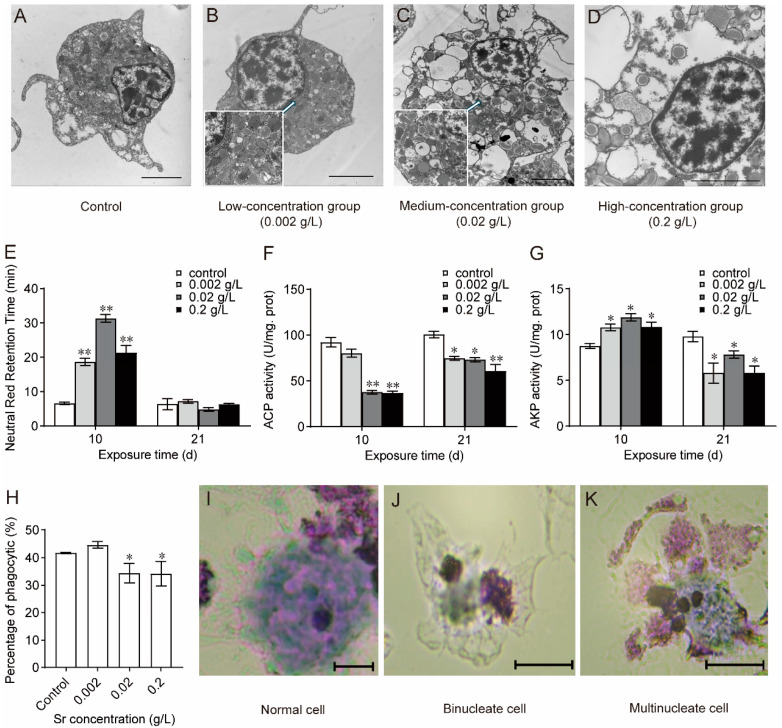
Sr-induced changes in the hemocyte structure and function of hemocytes (*n* = 5). (**A**–**D**): Transmission electron microscopy images of the changes induced by Sr in the granulocytes in each experimental group. Scale bars = 2 μm. (**E**): Neutral red retention time (NRRT); (**F**): Acid phosphatase (ACP); (**G**): Alkaline phosphatase (AKP); (**H**): Phagocytosis. * Significant differences at *p* < 0.05 level; ** Significant differences at *p* < 0.01 level. (**I**–**K**): The morphology of normal cells and MN cells. Bar: 50 μm.

**Figure 4 antioxidants-13-00464-f004:**
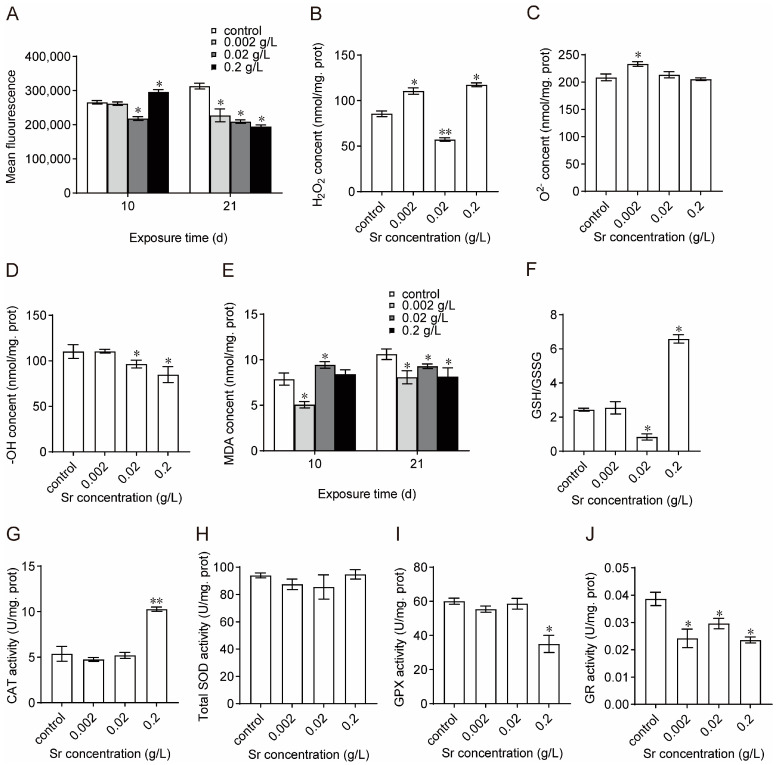
Changes in oxidation-reduction homeostasis and antioxidant system activity in *M. edulis* induced by Sr (*n* = 5). (**A**): ROS level; (**B**): H_2_O_2_ content; (**C**): O^2−^ content; (**D**): -OH content; (**E**): MDA content; (**F**): the ratio of GSH/GSSG; (**G**): CAT activity; (**H**): SOD activity; (**I**): GPX activity; (**J**): GR activity. * Significant differences at *p* < 0.05 level; ** Significant differences at *p* < 0.01 level.

**Figure 5 antioxidants-13-00464-f005:**
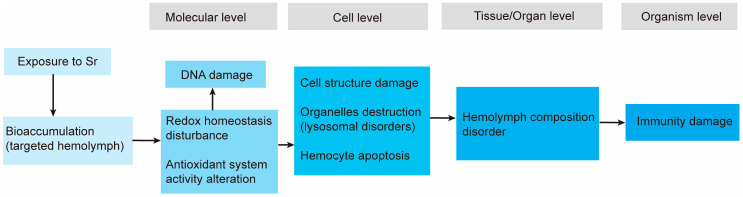
The adverse outcome pathway (AOP) of Sr with respect to *M. edulis*.

## Data Availability

Data is contained within the article and Appendix A.

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
