# Peer review of "Antioxidant Systematic Alteration Was Responsible for Injuries Inflicted on the Marine Blue Mussel Mytilus edulis Following Strontium Exposure"

_antioxidants, 2024, doi:10.3390/antiox13040464_

Round 1

Reviewer 1 Report

The work Using the adverse outcome pathway (AOP) framework to simu-2 late and evaluate the biological effects of nuclear-contaminated 3 water discharge on the marine blue mussel Mytilus edulis present and interesting and relevant study. Several parameters related with mussels health and which changes can compromise animals development and populations stability where presented and discussed.

Title: I would not use the concept of AOP in the title; in fact, the work contributes relevant data to construct one, but it does not evaluate all the levels that AOP encompasses.

The abstract section is not completed, please remove the instructions and add the information about the work.

Please ensure that the species name M. edulis is always written in italics, even when used in section titles.

It is unclear if the authors measured motion or only used it as an endpoint in the AOP model. In the case that they measured it, a description in the MM is needed.

Also, why do the authos considered this as an important organism level endpoint on the AOP? Since adult M edulis tend to be sessile.

Author Response

请参阅附件。

Reviewer 2 Report

The work concerns the analysis of the effect of strontium on mussels. Throughout the work, the authors, for some reason, use the term 88Sr to emphasize that they studied stable isotopes of strontium. What was the isotopic purity of the tested compound? Should this be specified in materials and methods? Didn't it contain other stable isotopes of this element?

If the authors believe that the effect of 90Sr can be ignored during the 21-day test (line 281), why do they mention "nuclear-contaminated water discharge" in the title of the work and not the effect of Sr ions on mussels? What is the time it takes for mussel immune cells to divide/form? If binucleated cells can be observed in the experiment (MN test), wouldn't beta radiation increase their number? The discussion (lines 330-350) should be improved.

The finding in Line 298 contradicts the results in Fig. 1, where low environmental concentrations of Sr are not bioaccumulated.

Strontium is an analogue of calcium and in vertebrates it is quickly deposited in bones and teeth. So why the statement (line 302) that it will be excreted and not incorporated into the mussels’ shell material?

What are the indications for the analysis of oxidative effects? Does Sr have a pro-oxidant effect? Why haven't biomarkers of Ca metabolism been tested? The discussion (lines 352-380) should be improved, and relevant works indicating the appropriateness of the selection of biomarkers should be cited.

The last part of the discussion (lines 381-) contains statements that are not supported by the results. The tested Sr concentrations influenced the level of biomarkers in different ways (sometimes the concentration of 0.2 and sometimes 0.02 g/L had a greater effect). Can the small DNA damage (of the order of permille) observed in the MN test cause: “cell structure damage” or “organelles destruction” in such a short time? Hence, the process presented in Fig. 5 is rather speculative. Especially since the authors did not observe "motion inhibition", so why do they assume that this will be the final effect?

Fig. 1 should rather show statistical differences between tissues (see line 300 in the discussion).

Line 386. LD50 is not an abbreviation of the half-lethal concentration. It is a dose! Exposure to LC50 should kill half of the mussels. I do not understand this sentence.

Reviewer 3 Report

The overall manuscript is quite decent, i pointed out some issues according to the experimental design and some images which can be improved 

Figure 3 (I,J,K): I suggest the authors to provide a higher quality figure, these are quite blurry.

Why the authors did not provide a figure showing the accumulation of ROS in fluorescence?

Round 2

Reviewer 2 Report

I accept all corrections made by the authors. 

However, I still believe that the use of the symbol 88Sr, both in text and in keywords, suggests the study of the influence of an isotope (radioisotope by default), even though this isotope is not radioactive. If the authors did not want to suggest research on the radioactive isotope, they would have simply written Sr.

I accept all corrections made by the authors. 
